# β-Xylosidase SRBX1 Activity from *Sporisorium reilianum* and Its Synergism with Xylanase SRXL1 in Xylose Release from Corn Hemicellulose

**DOI:** 10.3390/jof8121295

**Published:** 2022-12-13

**Authors:** Yuridia Mercado-Flores, Alejandro Téllez-Jurado, Carlos Iván Lopéz-Gil, Miguel Angel Anducho-Reyes

**Affiliations:** Dirección de Investigación, Innovación y Posgrado, Carretera Pachuca-Cd. Sahagún Km 20, Ex-Hacienda de Santa Bárbara, Universidad Politécnica de Pachuca, Zempoala 43830, Hidalgo, Mexico

**Keywords:** head smut, phytopathogen, glycosyl hydrolase, multifunctional enzymes

## Abstract

*Sposisorium reilianum* is the causal agent of corn ear smut disease. Eleven genes have been identified in its genome that code for enzymes that could constitute its hemicellulosic system, three of which have been associated with two Endo-β-1,4-xylanases and one with α-L-arabinofuranosidase activity. In this study, the native protein extracellular with β-xylosidase activity, called SRBX1, produced by this basidiomycete was analyzed by performing production kinetics and its subsequent purification by gel filtration. The enzyme was characterized biochemically and sequenced. Finally, its synergism with Xylanase SRXL1 was determined. Its activity was higher in a medium with corn hemicellulose and glucose as carbon sources. The purified protein was a monomer associated with the *sr16700* gene, with a molecular weight of 117 kDa and optimal activity at 60 °C in a pH range of 4–7, which had the ability to hydrolyze the ρ-nitrophenyl β-D-xylanopyranoside and ρ-Nitrophenyl α-L-arabinofuranoside substrates. Its activity was strongly inhibited by silver ions and presented *Km* and *Vmax* values of 2.5 mM and 0.2 μmol/min/mg, respectively, using ρ-nitrophenyl β-D-xylanopyranoside as a substrate. The enzyme degrades corn hemicellulose and birch xylan in combination and in sequential synergism with the xylanase SRXL1.

## 1. Introduction

Head smut is a worldwide disease caused by the dimorphic fungus *S. reilianum*, characterized by naked black sori with a powdery appearance in the ears and heads constituted by teliospore masses that fall on the soil and remain viable for up to 20 years. These structures can be spread in large agricultural areas by wind, rain, and irrigation, as well as by farm machinery and farmers themselves. The life cycle of this basidiomycete begins when the corn seeds germinate or during the first 15 days of seedling development. At that time, the teliospores produce a germinative tube, inside of which the nucleus is divided by meiosis to form four basidiospores haploids. This is considered the saprophytic phase of the fungus, which reproduces by gemmation. Plasmogamy occurs with the fusion of two sexually compatible yeasts, which produces the diploid infective mycelial that penetrates vegetal tissues to invade the plant. Its presence, however, is not manifested until flowering occurs with teliospore production in the plant’s reproductive organs. The losses that this pathogen causes to crops can reach 80%. Control methods for this phytosanitary problem include the use of tolerant hybrids and seed treatment with chemical or biological fungicides [1,2,3,4,5,6,7,8,9].

The plant cell wall is formed mainly of cellulose embedded in an amorphous matrix of hemicellulose and lignin. These structural polymers establish the first barrier against attacks by phytopathogens. Different saprophytic fungi produce extracellular enzymes that degrade the components of lignocellulosic plant tissues. The organisms then use the degradation products as assimilable substances for their development. In plant-pathogenic fungi, these activities are essential in plant cell wall depolymerization, which allows the microorganisms to penetrate and colonize their host. In addition, they generate nutrients that are consumed during pathogenic processes. In general, research has determined that phytopathogens have a greater number of genes that encode for these enzymes than saprophytic fungi of industrial importance, which indicates the value of these enzymatic activities in the life cycle of these organisms [10,11,12,13,14,15,16]

Of the polymers that form part of the plant cell wall, xylan is the most important component of hemicellulose. Its abundance is greater in angiosperms, where it reaches 15–30% of the dry weight, while in gymnosperms, it constitutes only 7–12%. It is a heteropolysaccharide of β-1,4-D-xylose units substituted with acetyl, L-arabinofuranosyl, galactosyl, glucuronyl, and 4-O-methylglucuronyl groups. The biological hydrolysis of this material requires the synergistic action of several enzymes, such as endoxylanase, β-xylosidases, arabinofuranosidases, acetyl xylan esterases, and glucuronidases. The first two play a crucial role in degradation, as endoxylanases cut β-1-4 glycosidic bonds, generating xylooligosaccharides that are used as a substrate by β-xylosidases, thereby generating xylose as a product [17,18,19,20,21,22].

In the *S. reilianum* genome sequence, the enzymes that may constitute its hemicellulolytic system have been identified. This system could be made up of two Endo-β-1,4-xylanases of the 10-family of glycoside hydrolases encoded by the *sr14403* and *sr15309* genes. There are three β-xylosidases of the 3-family (*sr16869*, *sr16700,* and *sr10116*) and one of the 43-family (*sr15773*), as well as three α-L-arabinofuranosidases of the 51-family (*sr12124*, *sr12911*, and *sr15761*), and two that correspond to the 54- (*sr12538*) and 62-families (*sr15193*). The endo-β-1,4-xylanase called SRXL1 has been associated with the *sr14403* gene, which has been purified and characterized biochemically. In addition, studies have demonstrated the presence of its intracellular isoform, SRXL1i. In another aspect, the overexpression of the products of *sr15309* (Endo-β-1,4-xylanase) and *sr15193* (α-L-arabinofuranosidase) has been observed in the secretomes of *S. reilianum* during the early infection of corn mesocotyls of an isogenic resistant line and another that is sensitive to head smut; this was associated with xylan degradation [23,24,25,26,27,28]. The enzymes that degrade the cell wall components produced by this basidiomycete could be attractive targets for designing disease-control strategies through specific enzyme inhibitors. They could be used for biotechnological applications to generate products based on the degradation of hemicellulosic materials.

Based on the above, the objective of this work was to contribute to the knowledge of the hemicellulolytic system of *S. reilianum* through the purification and biochemical characterization of a β-xylosidase to determine its synergism with endo-β-1,4-xylanase SRXL1 in corn cob hemicellulose hydrolysis.

## 2. Materials and Methods

### 2.1. Microorganism and Strain Conservation

A diploid strain isolated from maize crops in the state of Hidalgo, Mexico, was used in this study, which was donated by Dr. Santos Gerardo Leyva Mir at the Autonomous University of Chapingo, Mexico. For conservation, 48 h cultures incubated at 28 °C in inclined tubes with YEPD broth (1% yeast extract, 2% peptone, 2% glucose, and 2% agar) were prepared. The ensuing growth was covered entirely with sterile mineral oil and stored at room temperature [6].

### 2.2. Production of β-Xylosidase Activity

The production of β-xylosidase activity was determined in the cultures of *S. reilianum* in three different liquid media using a salt solution as a base (KH_2_PO_4_, 0.6 g/L; MgSO_4_–7H_2_O, 0.5 g/L; K_2_HPO_4_, 0.4 g/L; FeSO_4_–7H_2_O, 0.05 g/L; MnSO_4_–H_2_O, 0.05 g/L; and y ZnSO_4_–7H_2_O, 0.001 g/L) [25] supplemented with yeast extract at 5 g/L and the following carbon sources at 5 g/L each: glucose, corn hemicellulose, and glucose–corn hemicellulose. A pre-inoculum was prepared from a 24 h culture in YEPD broth incubated at 28 °C under agitation at 150 rpm. The resulting cells were collected by centrifugation at 5000 rpm, washed twice with sterile distilled water, and resuspended in the same medium in which they had been inoculated. Fifty-milliliter flasks with ten milliliters of each of the media described above were inoculated with this suspension, adjusting the absorbance to 0.2 at 660 nm in a Thermo Scientific Biomate 3 spectrophotometer. A flask of the medium was collected every 24 h, and the content was centrifuged to separate the biomass and collect the supernatant or crude enzyme extract (ECE), which was then used to determine the β-xylosidase activity. The biomass was determined by the increase in absorbance at 660 nm. 

Corn hemicellulose was obtained from corncobs by alkaline extraction and ethanol precipitation. The material was cut into fragments of approximately 4 cm, of which 60 g was taken and placed in 700 mL of boiling distilled water for 30 min. Then, solids were recovered and excess water was removed by drying in an oven at 70 °C for 18 h. After that, the material was mixed with NaOH at 12% and incubated for 12 h at room temperature. The mixture was then filtered. The filtrate (alkaline liquor) was recovered and mixed with ethanol in a 1:2 ratio. This mixture was incubated for 15 min at room temperature. The hemicellulose precipitated was collected by filtration and washed with ethanol to reach a pH value of 6 in the filtrate. Finally, the hemicellulose obtained was dried and analyzed using an Agilent Technologies Cary 630FT-IR (Fourier-transform infrared spectroscopy) spectrophotometer.

### 2.3. Enzymatic Assays

The β-xylosidase activity was determined by quantifying the release of ρ-nitrophenol using ρ-nitrophenyl β-D-xylanopyranoside as a substrate at 2 mM in 50 mM pH 4.5 acetate buffer. An amount of 50 µL of this solution was mixed with 50 µL of ECE. The mixture was incubated at 50 °C for 5 min; then, 900 µL of Na_2_CO_3_ at 0.1 M was added to stop the reaction. The ρ-nitrophenol released was determined at 405 nm using a standard curve. One unit of β-xylosidase activity was defined as the amount of enzyme that released one µmol of ρ-nitrophenol per minute under the assay conditions described.

### 2.4. β-Xylosidase SRBX1 Purification

The purified β-xylosidase was identified as SRBX1, where SR refers to *S. reilianum* and BX to β-xylosidase. The number 1 was assigned because it was the first purified enzyme to show this activity in this basidiomycete.

For purification, the ECE was obtained from a 316 h culture in a medium with hemicellulose–glucose that was filtered using a 0.45 µm pore membrane. The filtrate was passed through a Sephadex 10 12/300 gel filtration column coupled to an ÄKTA pure 25 L FPLC system. The proteins were eluted at a 1 mL/min flow rate with a 20 mM pH 4.0 acetate buffer. In each fraction, the enzymatic activity and protein concentration were determined as described above and by the Bradford method [29], respectively. The purification process was followed by SDS-PAGE. The gels were dyed with silver [30].

### 2.5. β-Xylosidase SRBX1 Characterization

The molecular weight of the purified protein was estimated by SDS-PAGE and gel filtration using mixtures of known proteins (BenchMark™ (Tempe, AZ, USA) Protein Ladder and BioRad (Hercules, CA, USA) Gel Filtration Standard, respectively).

The optimum temperature was determined by evaluating the activity of the pure enzyme in the range of 4–80 °C. Temperature stability was achieved by pre-incubating at 4, 25, 30, 35, 40, 45, 50, 55, 60, 70, and 80 °C for 1 h, followed by the enzymatic assay at the optimum temperature.

The optimal pH was determined at the optimal temperature using McIlvaine (pH 2–7), Tris-HCl (pH 7–10), and glycine-NaOH (pH 9–11.0) buffers at 100 mM. The pH stability was determined by pre-incubating the enzyme for 12 h at 4 °C in the presence of each buffer mentioned previously, followed by enzymatic determination.

The activity of the purified β-xylosidase SRBX1 was evaluated at the optimal pH and temperature on the chromogenic substrates ρ-Nitrophenyl α-L-arabinofuranoside, ρ-nitrophenyl β-D-glucopyranoside, and ρ-nitrophenyl β-D-xylanopyranoside, as well as on birch xylan [31].

The effects of metal ions and chemical agents on β-xylosidase SRBX1 were analyzed by preincubating the enzyme for 5 min at 60 °C in 2 and 10 mM solutions of Ca^2+^, Co^2+^, Ag^2+^, Cu^2+^, Fe^2+^, Mg^2+^, Mn^2+^, Zn^2+^, Na^+^, Li^+^, K^+^, EDTA (ethylenediaminetetraacetic acid), SDS (sodium dodecyl sulfate), and βME (β-mercaptoethanol), followed by the enzymatic assay. The results were expressed as a percentage of activity, taking as 100% the value obtained without adding to the reaction mixes of the ions or chemical compounds. The data obtained were compared by a two-way ANOVA, followed by Bonferroni’s post hoc test, where the concentrations and ions or chemical agents were evaluated as independent factors. The level of significance was set at *p* < 0.001.

The kinetic parameters of the purified enzyme, *Km* and *Vmax*, were determined using the Lineweaver–Burk method with ρ-nitrophenyl β-D-xylanopyranoside as the substrate at 1–7 mM in acetate buffer at pH 4 [32].

The SRBX1 protein was sequenced using the “Proteomics Discovery Platform” (Montreal Clinical Research Institute, MCRI) service by analyzing the protein identification with LC-MS/MS (Ion Trap). A database search was then performed against UniProt–*S. reilianum* and UniProt–Basidiomycota. 

The theoretical sequence of the purified proteins was analyzed in order to determine the possible glycosidic regions, pI and molecular mass, and the peptide signal and active sites using the Prosite database of protein domains, families, and functional sites (https://prosite.expasy.org, accessed on 5 November 2022), Peptide mass (https://www.expasy.org/resources/peptidemass, accessed on 5 November 2022), SignalP 4.1 (http://www.cbs.dtu.dk/services/SignalP/, accessed on 5 November 2022), and Pfam (https://pfam.xfam.org/, accessed on 5 November 2022), respectively. The structural modeling of β-xylosidase was performed using the bioinformatic webserver AlphaFold Colab (https://colab.research.google.com/github/deepmind/alphafold/blob/main/notebooks/AlphaFold.ipynb#scrollTo=XUo6foMQxwS2, accessed on 5 November 2022), as described by Jumper et al. [33].

### 2.6. Xylanase SRXL1 and β-Xylosidase SRBX1 Synergism in Hemicellulose Degradation

The synergism between the xylanase SRXL1 and β-xylosidase SRBX1 was evaluated using pure enzymes. Xylanase production and purification were conducted as described by Álvarez-Cervantes et al. [25].

Cob hemicellulose and birch xylan were utilized as the hydrolysis substrates. The latter was used to compare the degradation processes. Both polysaccharides were prepared at 0.5% in acetate buffer with pH 5.3. Treatments were carried out in 1 mL with 0.5 and 2.0 U of xylanase and β-xylosidase, respectively. Each one was incubated for 12 h at 50 °C. The reaction was stopped by placing the reaction mixture in a boiling water bath for 10 min.

For the sequential hydrolyses, the treatments were as follows: Sequence 1: a mixture of 0.5 U of xylanase with the substrate was incubated for 12 h at 50 °C. The reaction was stopped in a boiling water bath for 10 min. Once the mixture had cooled, 2.0 U of β-xylosidase was added, followed by incubation under the same conditions. The reaction was stopped by placing the reaction mixture in a bath of boiling water for 10 min. Sequence 2: a mixture of 2.0 U of β-xylosidase with the substrate was incubated for 12 h at 50 °C, and the reaction was stopped in a bath of boiling water for 10 min. Once the mixture had cooled, 0.5 U of xylanase was added, followed by incubation under the same conditions. The reaction was stopped by placing the reaction mixture in a bath of boiling water for 10 min. 

The effect of the concentrations of the xylanase/β-xylosidase enzymes during the hydrolysis of the substrates was evaluated using the following enzymatic unit ratios: 0.5/0.0, 1.0/0.0, 1.5/0.0, 2.0/0.0, 1.5/0.5, 1.0/1.0, 0.5/1.5, 0.0/2.0, 0.0/1.5, 0.0/1.0, and 0.0/0.5. The enzyme concentration, pH, temperature, and incubation time of the enzymatic reaction were identical to those outlined above.

In each case, a control was prepared using distilled water instead of the enzymes. 

The determination of the xylose released was carried out using a Thermo Scientific Ultimate 3000 HPLC. The samples were centrifuged at 15,000 rpm for 10 min and filtered with a 0.45 µm pore filter. They were then analyzed in a 300 × 7.7 mm 8 μm HyperREZ XP carbohydrate H column using water as the eluant at a flow rate of 0.6 mL/min. A xylose standard was used as the patron to determine the concentration in mg/mL. 

The degree of synergism was defined as the ratio of the xylose released through the action of the two enzymes divided by the sum of the xylose released through the action of each one according to the following equation [34]: $$\frac{XRXB}{\left(XRX\right)+\left(XRB\right)}$$
where *XRAB* is the xylose released by xylanase and β-xylosidase, *XRX* is the xylose released by xylanase action, and *XRB* is the xylose released by β-xylosidase action. 

In the experiment on the effect of the concentration of the xylanase/β-xylosidase during the hydrolysis of the substrates, the corresponding units of each enzyme were taken to determine the degree of synergism. All statistical analyses of the data obtained were performed using ANOVAs and Tukey’s test. 

## 3. Results

### 3.1. Production of β-Xylosidase Activity

Producing β-xylosidase required obtaining corn hemicellulose to be used as a carbon source in the culture media and in the synergism experiments. In this study, the polysaccharides were extracted from corn cobs as described in the Section 2. The FT-IR analysis of the extracted polymers showed the characteristic wavenumbers of hemicellulose obtained by the alkaline method (1385, 1329, and 1246 cm^−1^). Wavenumber 1044 cm^−1^ indicated the water absorbed into the hemicellulose, and 897 cm^−1^ referred to the glycosidic bonds. The wavenumbers attributed to O–H and C–H bonds were observed at 3419 and 2920 cm^−1^, respectively. The lignin monomers were identified at 1600 nm^−1^ (Appendix A). 

β-xylosidase production was determined in the culture supernatants in the different media. The highest activity was found in the medium with corn hemicellulose and glucose as the carbon sources (Figure 1). 

### 3.2. β-Xylosidase SRBX1 Purification and Characterization

A protein called SRBX1 with β-xylosidase activity was purified in one chromatographic step using a molecular exclusion column (Figure 2a). The protein had molecular weights of 117 and 107 kDa estimated by filtration gel and SDS-PAGE, respectively (Figure 2b). The enzyme was purified 592.6 times with a yield of 85.7% (Table 1). 

The purified enzymes were evaluated at different temperatures and pH values to determine the optimal conditions for catalysis and to define their stability under those same parameters. The optimum enzymatic activity occurred at 60 °C in a pH range of 4–7 (Figure 3a). After incubation at 4 °C for 12 h at a pH of 3–10, SRBX1 maintained activity above 76%. The activity decreased when incubating for 12 h at temperatures above 55 °C; however, in the range of 30–50 °C, optimal catalysis was maintained (Figure 3b). 

The β-xylosidase SRBX1 was evaluated against three other chromogenic substrates and birch xylan. The results showed that the enzyme had maximum activity against ρ-nitrophenyl β-D-xylanopiranoside, followed by ρ-Nitrophenyl α-L-arabinofuranoside. No activity against ρ-nitrophenyl β-D-glucopyranoside or birch xylan was observed (Table 2).

In all cases, adding different cations and chemical substances to the enzymatic reaction affected the β-xylosidase SRBX1 activity, with Ag^2+^ (2 and 10 mM), Cu^2+^ (10 mM), and βME (10 mM) showing the highest inhibitory effects. The data obtained presented statistically significant differences in relation to the control (Figure 4), except for Fe^2+^ and SDS at concentrations of 2 mM.

The kinetic parameters of β-xylosidase SRBX1 were determined using ρ-nitrophenyl β-D-xylanopiranoside as a substrate and the Lineweaver–Burk equation. The *Km* and *Vmax* values for the enzyme were 2.5 mM and 0.2 μmol/min/mg, respectively. 

The purified enzyme was sequenced, obtaining 28 peptides, which showed 100% similarity to the theoretical sequence of the protein that corresponded to the *sr16700* gene in the S. reilianum genome, which encoded for a β-xylosidase. The theoretical protein had 874 amino acids. The signal peptide was located from amino acids 1 to 23. It had 12 theoretical glycosylation sites and a molecular weight of 95.23 kDa. The sequence of amino acids from 322–340 showed that the protein belonged to the glycosyl hydrolases family. The active site was found in this region (Appendix A). 

Figure 5 shows the three-dimensional structure of the β-xylosidase SRBX1 protein encoded by the *sr16700* gene. Except for glycosylation site number 11, all other sites were found on the surface of the tertiary structure of the protein (Figure 5c). By signal peptide elimination, the glycosyl hydrolases family 3 motive was located on amino acids 301–318, as can be seen in Figure 5d,e. Finally, aspartate 315 (ASP315 or D315), involved in enzymatic catalysis, was observed (Figure 5e). 

### 3.3. Xylanase SRXL1 and β-Xylosidase SRBX1 Synergism in Hemicellulose Degradation

The synergism between the xylanase SRXL1 and the β-xylosidase SRBX1 in the xylose released from corn hemicellulose and birch xylan was evaluated. The mixture of the enzymes led to an increase in the released monosaccharide concentration. The synergism degree values obtained showed that it was possible to use both enzymes in a single step. The application order of each enzyme for degradation was specific, in which the xylanase acted first, followed by the β-xylosidase (Figure 6).

The effect of the mixture of the two enzymes at different activity units was evaluated in terms of xylose release and in the degree of synergism in the degradation of corn hemicellulose (Figure 7a) and birch xylan (Figure 7b). The findings showed that, when the same units of the two enzymatic activities were used (1.0 U/mL of xylanase SRXL1 and 1.0 of β-xylosidase SRBX1), xylose release was higher. This effect was also observed in the degree of synergism. Finally, when birch xylan was used as the substrate, the enzymatic activity of the mixture was more efficient. 

## 4. Discussion

The present study found that *S. reilianum* produced β-xylosidase activity in liquid culture, with corn hemicellulose acting as an inducer. In contrast, the levels of enzymatic activity were low in the medium with glucose. The expression of microbial enzymes that degrade hemicellulose increased when the microorganism grew in the presence of this material, so the available carbon source could be utilized. In this process, the monosaccharides that make up the polymer play essential roles as inducers that trigger the production and action of the entire depolymerization system, which is also subject to catabolic repression by glucose [35,36]. Interestingly, combining hemicellulose and glucose as carbon sources increased the production of β-xylosidase SRBX1 activity. This result is likely due to the presence of possible isoenzymes, given that five genes in the *S. reilianum* genome code for possible β-xylosidases, some of which may not respond to catabolic repression [23].

Fungal β-xylosidases have a wide variety of physicochemical and biological properties and specificity with respect to hemicellulose degradation. Most enzymes with this activity have molecular weights above 100 kDa, although they can be smaller, as well as monomeric or dimeric [20,35,36,37,38]. These proteins generally present their maximum catalytic activity at temperatures of 30–70 °C and at pH values ranging from 4.0 to 6.0. There are some activities that operate under acidic conditions. For example, *Penicillium sclerotiorum* β-xylosidase presents its maximum catalytic activity at pH 2.5, in contrast to that in *Sporotrichum thermophile* and *Talaromyces thermophilus*, which is more active at pH 7.0 [36,39,40,41]. The *S. reilianum* β-xylosidase SRBX1 purified in this work presented a molecular weight of 107–117 kDa and optimum activity at 60 °C in a pH range of 4–7, in addition to being monomeric. The enzyme may have potential applications in processes where the pH and temperature are conditioning factors during the hydrolysis of lignocellulosic residue to obtain fermentable sugars or xylitol [36].

The effect of some metals and chemical agents on the activity of the purified β-xylosidase was evaluated. All the agents tested decreased the enzymatic activity, while Cu^2+^ and β-mercaptoethanol at a concentration of 10 mM presented a strong inhibitory effect. The copper ion has been reported previously as a potent inhibitor of this type of enzymatic activity [42,43,44]. However, as with β-mercaptoethanol, the results can be highly variable. For example; Hayashi et al. [42] found a 3% decrease in beta-xylosidase activity from *Aureobasidium* at a concentration of 1 mM, while de Vargas et al. [43] reported that this same compound did not affect β-xylosidase activity produced by *Aspergillus versicolor*. The effect of β-mercaptoethanol on enzymes occurs because it reduces disulfide bonds, thus affecting their structure and, consequently, their activity [45]. In the case of β-xylosidase SRBX1, inhibition was likely due to structural modification through the reduction of disulfide bridges. The most significant inhibitory effect was found with silver ions. A similar effect has been found for β-xylosidase from the fungus *Neocallimastix frontalis*, where its activity decreased by 5.2% at a concentration of 1 mM [46].

The *Km* and *Vmax* values for the β-xylosidases of some fungi on the β-D-xylanopyranoside substrate can be variable. For example, the kinetic parameters for the enzyme purified in this work were 2.46 mM and 0.2 μmol/min/mg for *Km* and *Vmax*, respectively. For β-xylosidase from *Aspergillus versicolor* induced by xylose, the *Km* value was 0.32 mM, while that for another enzyme from this same fungus with the same catalytic activity, but induced by xylan, was 0.19 mM [43]. The values determined for the enzyme produced by *Fusarium proliferatum* were 0.77 mM and 75 μmol/min/mg [37], while for the β-xylosidase of *Fusarium verticilloides*, the *Km* value was 0.85 mM [47]. In their work, Kim and Yoon [48] reported *Km* and *Vmax* values of 1.1 mM and 1.4 μmol/min/mg, respectively, for β-xylosidase/arabinofuranosidase from *Paenibacillus woosongensis* expressed in *E. coli*, whereas the values for the heterologous β-xylosidase from *Aspergillus oryzae* were 0.48 mM and 42.6 μmol/min/mg, respectively [38].

The sequencing of the protein revealed its association with the *sr16700* gene, which codes for a β-xylosidase [23]. Activity assays using three different substrates allowed us to identify that this enzyme had a higher activity of β-xylosidase, followed by a lesser degree of activity of α-L-arabinofuranosidase. This protein belongs to family 3 of the glycosyl hydrolases, many of which have been identified as bi/multifunctional on synthetic substrates [49]. Some microorganisms produce these enzymes with the ability to hydrolyze various substrates, which are characterized by catalytic domains that perform various functions [50]. It has been suggested that the relaxed specificity of some enzymes may be an essential aspect that demonstrates an evolutionary advantage when using existing proteins for another function with no need to expand the genome. In addition, this is a tool with multiple industrial applications [51,52].

β-xylosidase SRBX1 corresponds to a secreted monomeric protein with a theoretical molecular weight of 95.23 kDa; however, the weight obtained experimentally by gel filtration was 117 kDa. This suggests that the enzyme may be glycosylated, as has been reported for these types of activities [53]. Therefore, the finding mentioned above can be attributed to the presence of 12 possible glycosylation sites, 11 of which were found on the surface of the protein. Furthermore, it is well known that, during post-translational processing, 2.5 kDa is added per N-glycosylation site [54]. This could explain the difference in size between the theoretical and purified proteins.

Studies have shown that the biological hydrolysis of the hemicellulose present in lignocellulose requires the combination of several glycosyl hydrolases, where different activities generate a synergistic effect for the complete depolymerization of the substrate. Understanding the characteristics and cooperation mechanisms of these enzymes is essential for determining how synergism occurs, as it may be sequential or simultaneous. In most cases, endoxylanases act first, releasing xylooligosaccharides that are the substrates for the β-xylosidases [17,18,19,36,55,56]. In the present study, the synergism of the xylanase SRXL1 and the β-xylosidase SRBX1 produced by *S. reilianum* during the hydrolysis of corn cob hemicellulose and birch xylan was evaluated. The highest degree of synergism was found when the two enzymes were mixed and when they were used sequentially (xylanase first, then β-xylosidase). This process proved to be more efficient on birch xylan, likely because this substrate was a commercial product. Yang et al. [21] evaluated different simultaneous reactions with xylanase (Xyn11A) and two enzymes with β-xylosidase activity, α-arabinoside and xylanase (Xyl43A and Xyl43B), from the fungus *Humicola insolens*. The enzymatic combination of Xyn11A and Xyl43A released more reducing sugars from birch xylan and beech xylan, both of which were commercial products. In contrast, the Xyn11A–Xyl43B mixture was the best for commercial wheat arabinoxylan. In all cases, the most significant synergy was found when the substrate was incubated first with xylanase Xyn11A, with Xyn11A and Xyl43A being added later. In a study by Tuncer and Ball [57], oat xylan was hydrolyzed using a xylanase, a β-xylosidase, and an arabinofuranosidase produced by the actinomycete *Thermomonospora fusca*. The individual enzymes released 2.8, 0.46, and 0.7 mg/mL of xylose, respectively. When used together, however, 5.9 mg/mL of the monosaccharide was obtained. Zheng et al. [58] reported high levels of synergism (12.02) when the acidophilic β-xylanase Xyl11 from *Trichoderma asperellum* was used in combination with a commercial β-xylosidase.

In recent years, the study of enzymes that degrade lignocellulosic materials to obtain industrial products has gained importance in developing eco-friendly processes. In this field of research, phytopathogenic fungi could become sources of new enzymes with biotechnological potential [13,59]. However, studying these activities from a phytopathological perspective is essential because they could also be important virulence factors, as has been described for Arabinofuranosidase/β-xylosidase *Sclerotinia sclerotiorum*, where null mutants of this activity considerably reduced its capacity to cause necrosis in canola stems and leaves [60]. This fungus could use the combination of β-xylosidase SRBX1 with the xylanase SRXL1 during the infection and colonization processes by degrading the cell walls of plants. Moreover, there are potential applications for the degradation of hemicellulosic residues from corn to obtain products of industrial interest.

## 5. Conclusions

In this study, a protein with β-xylosidase activity produced by the fungus *S. reilianum* was purified and associated with the *sr16700* gene. The enzyme corresponded to an extracellular monomeric glycosyl hydrolase that belonged to family 3, with optimal activity in a wide range of pH (4–7) and at a temperature of 60 °C, which acted in simultaneous and sequential synergism with the xylanase SRXL1 from this same fungus, in the degradation of corn hemicellulose and birch xylan. The study of these enzymes contributes to the knowledge of the hemicellulolytic system of this basidiomycete, which could have implications for pathogenesis and possible industrial applications.

## Figures and Tables

**Figure 1 jof-08-01295-f001:**
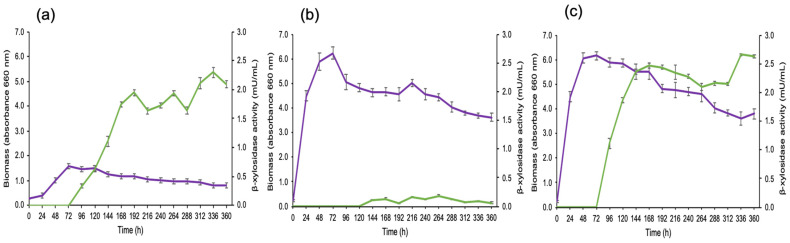
β-xylosidase production from *S. reilianum* in different culture media. (**a**) Minimum medium with glucose; (**b**) minimum medium with corn hemicellulose; and (**c**) minimum medium with corn hemicellulose and glucose. Green line: biomass determined by the increase in the absorbance at 660 nm. Purple line: β-xylosidase activity.

**Figure 2 jof-08-01295-f002:**
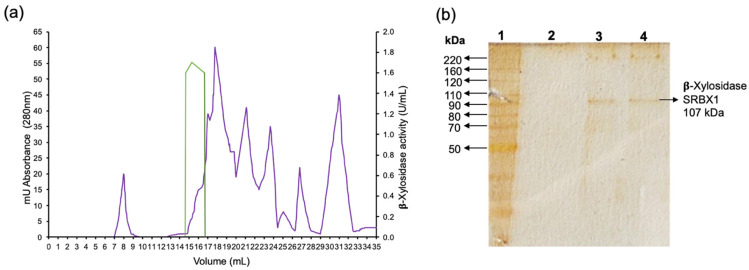
Purification of β-xylosidase SRBX1 from *S. reilianum*. (**a**) Chromatographic step in the gel filtration column. Green line: enzymatic activity. Purple line: absorbance at 260 nm. (**b**) Silver-stained SDS-PAGE. Line 1: molecular weight marker; line 2: without sample; lines 3 and 4: fractions with activity obtained by gel filtration chromatography.

**Figure 3 jof-08-01295-f003:**
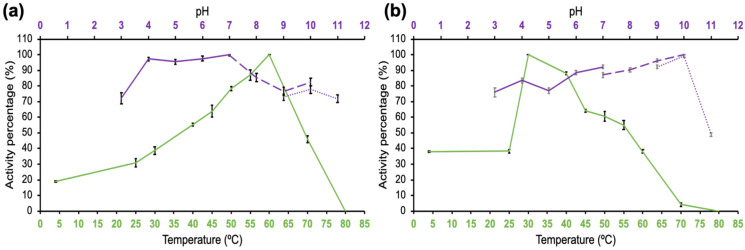
Effects of temperature and pH on β-xylosidase SRBX1 from *S. reilianum*. (**a**) Optimal temperature and pH. (**b**) Temperature and pH stability. Green line: temperature. Purple line: pH. Solid line: McIlvaine buffer (pH 2–7); dashed line: Tris–HCl buffer (pH 7–10); and dotted line: glycine–NaOH buffer (pH 9–11.0).

**Figure 4 jof-08-01295-f004:**
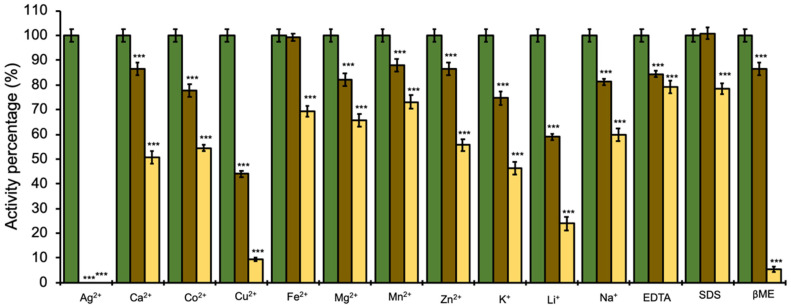
Effect of metal ions and chemical substances on the activity of β-xylosidase SRBX1 from *S. reilianum*: green: 0 mM; brown: 2 mM; and yellow: 10 mM. *** *p* < 0.001. EDTA: ethylenediaminetetraacetic acid; SDS: sodium dodecyl sulfate; and βME: β-mercaptoethanol.

**Figure 5 jof-08-01295-f005:**
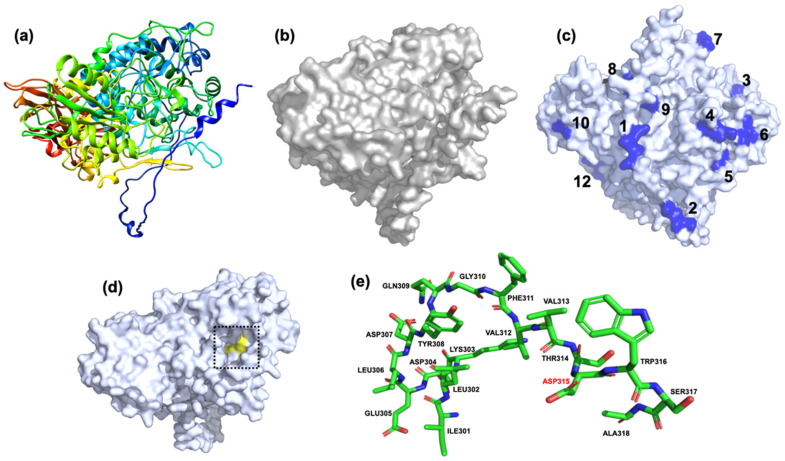
Structural modeling of the β-xylosidase SRBX1 protein from S. reilianum. (**a**,**b**) Three-dimensional structures. (**c**) Location of the glycosylation sites. The protein has 12 possible glycosylation sites. Except for site number 11, all others are located on the protein’s surface (indicated in blue). (**d**) The location of the active site is in yellow in the dashed box. (**e**) Amino acids of the active site. ASP315, which theoretically participates in enzymatic catalysis, is shown in the letters red. The structural modeling was performed using the bioinformatic web-server AlphaFold Colab (https://colab.research.google.com/github/deepmind/alphafold/blob/main/notebooks/AlphaFold.ipynb#scrollTo=XUo6foMQxwS2, accessed on 5 November 2022).

**Figure 6 jof-08-01295-f006:**
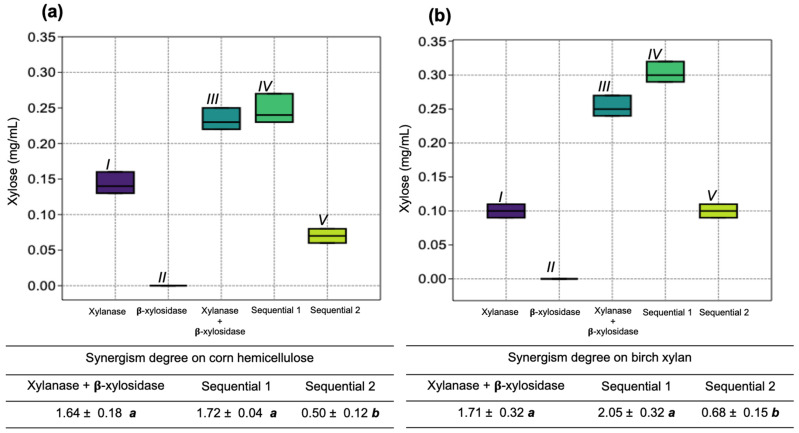
Xylose release and degree of synergism between xylanase SRXL1 and β-xylosidase SRBX1 activity from *S. reilianum* in the degradation of corn hemicellulose (**a**) and birch xylan (**b**). Sequence 1: xylanase was added first, followed by β-xylosidase. Sequence 2: β-xylosidase was added first, followed by xylanase. Amounts of 0.5 and 2.0 U of xylanase and the β-xylosidase were used, respectively. The same numbers roman indicates that no statistically significant difference was observed; *p* < 0.05.

**Figure 7 jof-08-01295-f007:**
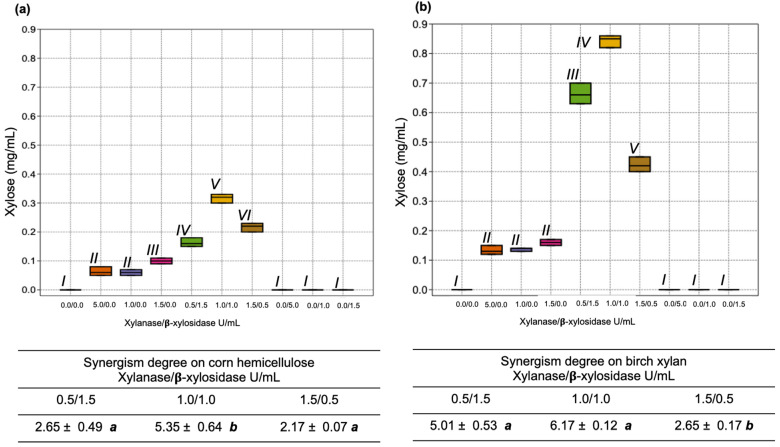
Xylose release and degree of synergism between the actions of xylanase SRXL1 and β-xylosidase SRBX1 from *S. reilianum* using different enzymatic units in the degradation of corn hemicellulose (**a**) and birch xylan (**b**). The same numbers roman indicates indicate that no statistically significant difference was observed; *p* < 0.05.

**Table 1 jof-08-01295-t001:** Purification of β-xylosidase SRBX1 from *S. reilianum*.

Purification Step	Volume (mL)	Total Protein (mg)	Total Activity (mU)	Specific Activity (mU/mg)	Yield (%)	Purification (Fold)
Crude extract	1	209	5.6	0.027	100	1
Gel filtration	2	0.3	4.8	16	85.7	592.6

**Table 2 jof-08-01295-t002:** Activity of β-xylosidase SRBX1 from *S. reilianum* against different substrates.

Substrate	Enzymatic Activity (%)
ρ-nitrophenyl β-D-xylanopiranoside	100.0 ± 0.0
ρ-Nitrophenyl α-L-arabinofuranoside	15.5 ± 0.8
ρ-nitrophenyl β-D-glucopyranoside	0.0 ± 0.0
Birch xylan	0.0 ± 0.0

## Data Availability

Not applicable.

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
