# Peer review of "β-Xylosidase SRBX1 Activity from Sporisorium reilianum and Its Synergism with Xylanase SRXL1 in Xylose Release from Corn Hemicellulose"

_jof, 2022, doi:10.3390/jof8121295_

Round 1

Reviewer 1 Report

The authors Mercado-Flores and colleges identified a protein fraction with beta xylosidase activity in cell extracts of the maize pathogenic fungus Sporisorium reilianum grown on maize hemicellulose and glucose. The identified the cognate protein and characterized its activity using biochemical activity assays. Last, they describe a synergistic action with a previously characterized xylanase SRXL1.

The presented work is scientifically sound and can be interesting for industrial application. I favour its publication with minimal changes (see below):

Abstract: lacks the information that this is work with the native protein

Introduction:

63ff: what family? Glycoside hydrolase?  

74ff: Sentence incomplete? Specific disease control difficult since xylanases present in many organisms.

Results:

Figure 1: it would be nice to have the cell count in the figure

Figures 2a and 3: green and brown lines hard to discriminate on my computer

Figure 5: mention alphafold

Author Response

Response to reviewer

In response to the comments and suggestions regarding the manuscript entitled “β-xylosidase SRBX1 activity from Sporisorium reilianum and its synergism with xylanase SRXL1 in the xylose release from corn hemicellulose” submitted to the Journal of Fungi, the following aspects are discussed:

Comment:

Abstract: lacks the information that this is work with the native protein

Reply:

In the abstract, the information that this is work with the native protein was included. Line 13.

Comment:

Introduction:

63ff: what family? Glycoside hydrolase? 

Reply:

In the introduction in line 65, the Glycoside hydrolase family was included.

Comment:

74ff: Sentence incomplete? Specific disease control difficult since xylanases present in many organisms.

Reply:

In the introduction, in lines 75 to 78, the sentence was modified for clarity.

It has been shown that plants produce enzyme inhibitors as a defense mechanism against attack by phytopathogens (Juge N. 2006. Plant protein inhibitors of cell wall degrading enzymes. Trends Plant Sci. 11: 359-367. https://doi .org/10.1016/j.tplants.2006.05.006), so we consider that it is possible to propose these enzymes as targets for the design of control strategies due these are involved in the plant pathogens penetration and colonization processes.

Comment:

Results:

Figure 1: it would be nice to have the cell count in the figure

Reply:

In Figure 1, the biomass quantification was incorporated into the graphics.

Comment:

Figures 2a and 3: green and brown lines hard to discriminate on my computer

Reply:

In Figures 2a and 3, the lines colors were changed for clarity.

Comment:

Figure 5: mention alphafold

Reply:

In Figure 5, the bioinformatic web-server AlphaFold Colab was mentioned.

All changes were highlighted in yellow.

Your comments and suggestions will contribute significantly to improving this manuscript.

Thank you for your observations,

Dr. Yuridia Mercado-Flores

Reviewer 2 Report

The present research article presents interesting results regarding the effect of β-xylosidase SRBX1 extracted from Sporisorium reilianum, on the xylose release from hemicellulose. Some sections of the manuscript have to be remade, such as the conclusion section which is not exactly a conclusion of the present work. The references section especially from the introduction should be updated, as the majority of the references are older than 5 years.

Some minor corrections:

-          line 18 – please insert a space between the number and degree sign, revise the whole manuscript

-          section between lines 27-41 –  needs more references, especially in the first statements (i.e. https://doi.org/10.1016/j.cj.2015.02.001). – the same applies to the rest of the introduction.

-          line 88 – please insert after YEPD the term broth

-          line 100 – please use fifteen or 50

-          line 101 – the absorbance was adjusted with the instrument?

-          line 114 – please specify the model and producer of the FT-IR instrument

-          every instrument has to be specified, the model, type, producer, and country! Please revise the whole material and method section. (i.e. 129-130 ÄKTA pure 129 FPLC system, 200 - HPLC).

-          line 155 – 158 – the statistical analyses were performed with the software? I suggest separating the statistical analyses in the material and method section with a separate subtitle, and also specifying the number of duplicates/triplicates performed

-          figure 3 – which is the red or the brown line, respectively?

-          line 274 – please correct gen > gene

-          line 318 – 321 – is mostly introduction not discussion

-          line 432 – please correct whit > with

-          please revise the conclusion section. First, there have to be no references included here.

This is not the conclusion of the present study! Revise the whole section and rewrite it accordingly!

After a thorough revision, the manuscript can be considered for publication.

Author Response

Response to reviewer

In response to the comments and suggestions regarding the manuscript entitled “β-xylosidase SRBX1 activity from Sporisorium reilianum and its synergism with xylanase SRXL1 in the xylose release from corn hemicellulose” submitted to the Journal of Fungi, the following aspects are discussed, and the changes were highlighted in blue.

Comment:

The present research article presents interesting results regarding the effect of β-xylosidase SRBX1 extracted from Sporisorium reilianum, on the xylose release from hemicellulose. Some sections of the manuscript have to be remade, such as the conclusion section which is not exactly a conclusion of the present work. The references section especially from the introduction should be updated, as the majority of the references are older than 5 years.

Reply:

The manuscript was reviewed, and the conclusion section was rewritten.

We conduct a thorough bibliographic search to select the most current references concerning the subject of the work presented; therefore, we consider that the references cited in the manuscript are current.

Comment:

line 18 – please insert a space between the number and degree sign, revise the whole manuscript

Reply:

The correction was made. Line 18.

Comment:

section between lines 27-41 –  needs more references, especially in the first statements (i.e. https://doi.org/10.1016/j.cj.2015.02.001). – the same applies to the rest of the introduction.

Reply:

More references were incorporated in the section from lines 27 to 41, the same as in the rest of the introduction.

Comment:

line 88 – please insert after YEPD the term broth

Reply:

The term broth was inserted. Line 90.

Comment:

line 100 – please use fifteen or 50

Reply:

50-mL was use. Line 103

Comment:

line 101 – the absorbance was adjusted with the instrument?

Reply:

The absorbance was adjusted with a spectrophotometer thermo scientific model Biomate 3. Lines 104-105.

Comment:

line 114 – please specify the model and producer of the FT-IR instrument

Reply:

The model and producer of the FT-IR instrument was specified. Lines 118-119.

Comment:

every instrument has to be specified, the model, type, producer, and country! Please revise the whole material and method section. (i.e. 129-130 ÄKTA pure 129 FPLC system, 200 - HPLC).

Reply:

For every instrument the model and producer was specified. Lines 104-105, 118-119 and 136-137.

Comment:

line 155 – 158 – the statistical analyses were performed with the software? I suggest separating the statistical analyses in the material and method section with a separate subtitle, and also specifying the number of duplicates/triplicates performed

Reply:

The statistical analyses were performed with the conventional methods that are mentioned in the manuscript. We not considered to separate in a section the statistical analyses because this could be confusing. 

Comment:

figure 3 – which is the red or the brown line, respectively?

Reply:

In the Figure 3 the colors in the lines were changed. Line 289.

Comment:

line 274 – please correct gen > gene

Reply:

In the line 306 the word gen was changed by gene.

Comment:

line 318 – 321 – is mostly introduction not discussion

Reply:

This section was eliminated in the discission.

Comment:

line 432 – please correct whit > with

Reply:

In the line 485 the word whit was changed by with.

Comment:

please revise the conclusion section. First, there have to be no references included here.

This is not the conclusion of the present study! Revise the whole section and rewrite it accordingly!

Reply:

The conclusion section was rewriter.

Your comments and suggestions will contribute significantly to improving this manuscript.

Thank you for your observations,

Dr. Yuridia Mercado-Flores

Round 2

Reviewer 2 Report

The manuscript has been corrected quite well; some minor remarks:

- please insert a space between the number and degree sign - revise the whole manuscript

- at the spectrophotometer Thermo 102 scientific model Biomate 3 - please specify the country of production - the same applies for all the utilised instruments

after these corrections the manuscript can be considered for publication